# The Vaginally Exposed Extracellular Vesicle of *Gardnerella vaginalis* Induces RANK/RANKL-Involved Systemic Inflammation in Mice

**DOI:** 10.3390/microorganisms13040955

**Published:** 2025-04-21

**Authors:** Yoon-Jung Shin, Xiaoyang Ma, Ji-Su Baek, Dong-Hyun Kim

**Affiliations:** Neurobiota Research Center, College of Pharmacy, Kyung Hee University, Seoul 02447, Republic of Korea; nayo971111@naver.com (Y.-J.S.); xiaoyangma12@gmail.com (X.M.); bjisoo500@naver.com (J.-S.B.)

**Keywords:** *Gardnerella vaginalis*, extracellular vesicle, vaginitis, osteoporosis, psychiatric disorder

## Abstract

*Gardnerella vaginalis* (GV), an opportunistic pathogen excessively proliferated in vaginal dysbiosis, causes systemic inflammation including vaginitis, neuroinflammation, and osteitis. To understand its systemic inflammation-triggering factor, we purified extracellular vesicles isolated from GV (gEVs) and examined their effect on the occurrence of vaginitis, osteitis, and neuroinflammation in mice with and without ovariectomy (Ov). The gEVs consisted of lipopolysaccharide, proteins, and nucleic acid and induced TNF-α and RANKL expression in macrophage cells. When the gEVs were vaginally exposed in mice without Ov, they significantly induced RANK, RANKL, and TNF-α expression and NF-κB^+^ cell numbers in the vagina, femur, hypothalamus, and hippocampus, as observed in GV infection. The gEVs decreased time spent in the open field (OT) in the elevated plus maze test by 47.3%, as well as the distance traveled in the central area (DC) by 28.6%. In the open field test, they also decreased the time spent in the central area (TC) by 39.3%. Additionally, gEVs decreased spontaneous alteration (SA) in the Y-maze test by 33.8% and the recognition index (RI) in the novel object recognition test by 26.5%, while increasing the immobility time (IT) in the tail suspension test by 36.7%. In mice with OV (Ov), the gEVs also induced RANK, RANKL, and TNF-α expression and increased NF-κB^+^ cell numbers in the vagina, femur, hypothalamus, and hippocampus compared to vehicle-treated mice. When gEVs were exposed to mice with Ov, gEVs also reduced the DC, TC, OT, SA, and RI to 62.1%, 62.7%, 28.2%, 90.7%, and 85.4% of mice with Ov, respectively, and increased IT to 122.9% of mice with Ov. Vaginally exposed fluorescein-isothiocyanate-tagged gEVs were detected in the blood, femur, and hippocampus. These findings indicate that GV-derived gEVs may induce systemic inflammation through the activation of RANK/RANKL-involved NF-κB signaling, leading to systemic disorders including vaginitis, osteoporosis, depression, and cognitive impairment. Therefore, gEVs may be an important risk factor for vaginitis, osteoporosis, depression, and cognitive impairment in women.

## 1. Introduction

A healthy vaginal microbiota consists of huge microbes including *Lactobacilli*, *Actinobacteria*, *Prevotella*, *Veillonella*, and *Streptococci* [1]. However, the composition of the vaginal microbiota fluctuates, influenced by both intrinsic and extrinsic factors, including menopause, the menstrual cycle, contraceptive use, and stress, leading to vaginal dysbiosis. Pathogens excessively proliferating in vaginal dysbiosis, such as the overgrowth of *Gardnerella vaginalis* (GV) and *Candida albicans*, causes vaginitis, including bacterial vaginosis [2,3]. Many patients with bacterial vaginosis (vaginitis) show increased levels of inflammatory cytokines, including tumor necrosis factor (TNF)-α.

Gut dysbiosis, which is caused by stresses such as pathogen infection or immobilization stress, induces depression and anxiety with systemic inflammation and gut inflammation [4,5]. Interestingly, the prevalence of depression is approximately two times higher in women than in men [6,7,8,9]. Many patients with depression show increased levels of inflammatory cytokines, including TNF-α and interleukin (IL)-6 [10]. These proinflammatory cytokines have been associated with depression and osteoporosis [11,12,13]. These findings suggest that vaginal dysbiosis-involved vaginitis may be associated with the occurrence of depression and osteoporosis.

*Gardnerella vaginalis* (GV), an opportunistic pathogen in the vagina, causes vaginitis-, osteoporosis-, and depression-like symptoms through the activation of receptor activator of nuclear factor κB (RANK)/RANK ligand (RANKL)- and TNF/TNF receptor (TNFR)-mediated NF-κB signaling [14,15,16]. However, the systemic inflammation-triggering factor of GV remains poorly understood.

Bacterial extracellular vesicles (EVs) have diverse biological effects [17,18]. The EVs from *Porphyromonas gingivalis* cause periodontitis and cognitive deficits in mice [19]. The EVs from *Paenalcaligenes hominis* and those from *Escherichia fergusonii* are linked to depression and cognitive impairment (DCi)-like symptoms with neuroinflammation in mice [20,21]. However, the role of GV-derived EVs (gEVs) in the trigger of systemic inflammation remains unclear.

Therefore, to explore the triggering factor of GV-induced systemic inflammation, we isolated gEVs from GV and investigated their pathogenesis in mice with or without ovariectomy (OV).

## 2. Materials and Methods

### 2.1. GV Culture and gEV Preparation

GV KCTC5096 was cultured anaerobically in brain heart infusion (BHI, BD, Franklin Lakes, NJ, USA) broth with 10% fetal bovine serum (FBS, Gibco, Frederick, MD, USA) and 1% yeast extract (Kerry, Ireland) at 37 °C for 48 h, as previously reported [22]. Bacterial cells were centrifuged at 8000× *g* for 20 min at 4 °C. Precipitated cells were suspended in sterilized saline for further experiments.

The gEVs (size, 30–80 nm) were purified from the supernatant of the GV culture using ultracentrifugation and an exoEasy Maxi Kit (Qiagen, Hilden, Germany), as described previously [21]. Briefly, the GV (1 L) was anaerobically cultured in BHI broth containing 5% FBS and 1% yeast extract at 37 °C for 48 h and centrifuged at 10,000× *g* for 20 min at 4 °C. The supernatant was filtered sequentially through 0.45 μm and 0.22 μm microporous filters and centrifuged at 120,000× *g* for 3 h at 4 °C. The precipitate was resuspended in saline. The gEVs were purified using ultracentrifugation and an exoEasy Maxi Kit (Qiagen, Hilden, Germany). The purified EVs were suspended in phosphate-buffered saline, added onto a copper-coated grid, washed with deionized water, negatively stained with 2% uranyl acetate, and observed under a transmission electron microscope (JEOL, Tokyo, Japan).

The freeze-dried gEVs contained 44.21% protein, 0.23% lipopolysaccharides (LPS), and 0.02% nucleic acids (DNA, 15.50%; RNA, 84.51%). The protein concentration was determined using the Bradford protein assay kit. The nucleic acid content was quantified using DNA and RNA purification kits (Qiagen), with the total amount calculated as the sum of DNA and RNA. Protein analysis of gEV was performed by SDS-PAGE and LC-MS/MS using a Thermo Fisher Orbitrap XL with Easy nLC 1000 [20]. Fluorescein isothiocyanate (FITC)-labeled gEVs (F-gEV: FITC-to-protein ratio, 0.01) were prepared using an FITC labeling kit (Sigma, St. Louis, MO, USA) according to the manufacturer’s protocol [21].

### 2.2. Animals and Housing

Female C57BL/6 mice (8 weeks old, 20–22 g) were purchased from Koatech Experimental Animal Breeding (Seoul, Republic of Korea). They were housed in standard cages under controlled conditions and fed commercial chow and water ad libitum. Mice were acclimated in the animal facility for one week before being used in the experiments.

All animal experiments were approved by the Kyung Hee University Laboratory Animal Care and Use Committee [IACUC No. KHUASP(SE)-21-591] and conducted according to the University and ARRIVE guidelines for laboratory animal care and use [23].

### 2.3. Macrophages

Macrophage cells were prepared as previously described [24]. The cells (1 × 10^6^ cells per well) were incubated with GV (1 × 10^4^ and 1 × 10^6^ CFU/mL) or gEVs (25, 200, and 400 ng/mL) for 24 h in DMEM medium (*n* = 4). TNF-α and IL-6 levels in the supernatant were measured using enzyme-linked immunosorbent assay (ELISA) kits, while RANK and RANKL levels in the cell lysate were analyzed using quantitative real-time polymerase chain reaction (qPCR).

### 2.4. Exposure to GV or gEV in the Vagina of Mice

To examine whether vaginal exposure to GV or gEVs could induce systemic inflammation, including vaginitis, they were applied to the vaginas of mice during the diestrus phase. Each experimental group included six mice.

Experiment 1—To investigate whether gEVs could induce systemic inflammation, mice were divided into three groups: Nc, Gv, and Ev. GV (1 × 10^6^ CFU/0.02 mL) and gEVs (10 µg protein/0.02 mL saline) were vaginally applied to Gv and Ev groups daily for 5 days, respectively. The Nc group received saline alone.Experiment 2—To examine the impact of OV on gEV-induced systemic inflammation, mice were divided into four groups: Sham (Sh), Ov, oGv (OV + GV), and oEv (OV + gEV). The Sh group was operated on without the resection of bilateral ovaries. All mice other than the Sh group underwent a bilateral ovary removal operation under 2.5% isoflurane, while Sh mice underwent an operation without ovary removal. GV (1 × 10^6^ CFU/0.02 mL) and gEVs (10 µg protein/0.02 mL saline) were vaginally applied to oGv and oEv groups daily for 5 days, respectively. Sh and Ov mice received sterilized saline.Experiment 3—To explore the translocated organs of vaginally applied gEVs, mice were divided into three groups: Nc, Ft, and fEv. F-gEV (2 µg/0.02 mL saline/day) was vaginally applied to the fEv mice daily for 3 days. Nc and Ft groups received normal saline and FITC (0.02 mL/day), respectively.

Behavioral assessments of DCi were conducted after the final exposure to GV or gEVs. Mice were euthanized by exposure to carbon dioxide in a chamber 20 h after completing the behavioral tests, followed by cervical dislocation. Plasma, vagina, uterus, and brains were removed and stored at −80 °C for further experiments. Plasma was prepared as previously reported [22].

### 2.5. Behavioral Tasks

Depression and anxiety-like behaviors were assessed utilizing the open field test (OFT), elevated plus maze task (EPMT), and tail suspension test (TST) [21,25]. Cognitive impairment-like behaviors were assessed utilizing the Y-maze test (YMT) and the novel object recognition test (NORT) [26].

### 2.6. ELISA and qPCR

The levels of cytokines TNF-α, IL-6, RANKL, serotonin, and estradiol in the hypothalamus, hippocampus, femur, vagina, and blood were determined using ELISA kits (TNF-α, IL-6, and RANKL assay kits, R&D Systems, Minneapolis, MN, USA; serotonin assay kit, Abcam, Cambridge, MA, USA; estradiol assay kit, Thermo Fisher Scientific, Waltham, MA, USA), as previously described [21]. RANK, RANKL, TNF-α, and IL-6 mRNA levels were assayed by qPCR, as previously described [21]. The sequences of the primers used are described in Appendix A.

### 2.7. Immunofluorescence, Hematoxylin and Eosin (H&E), and Tartrate-Resistant Acid Phosphatase (TRAP)/Alkaline Phosphatase (ALP) Staining

Immunofluorescence staining for vagina, femur, and brain tissues was prepared as previously reported [26]. Briefly, the tissue sections (25 μm thickness) were washed with PBS, incubated in citrate buffer (pH 6.0) at 95 °C for 10 min, blocked in 10% normal donkey serum (Abcam) for 1 h, incubated for 12 h at 4 °C with the primary antibodies (Iba1 [1:800, Abcam], NF-κB [1:800, Cell Signaling Technology, Danvers, MA, USA], TNF-α [1:200, Abcam], and RANK [1:250, Abcam]), washed with saline, and then incubated with Alexa (Alexa Fluor 594 or Alexa Fluor 488 [1:200, Invitrogen, Waltham, MA, USA]) for 2 h. Nuclei were stained with DAPI. Immunostained sections were observed using a confocal microscope.

H&E and immunofluorescence staining for vagina, femur, and brain were performed using an H&E staining kit (Agilent Dako, Santa Clara, CA, USA) according to the manufacturer’s instructions [22]. Femurs were stained using a TRACP/ALP staining kit (FUJIFILM, Tokyo, Japan) according to the manufacturer’s protocol.

### 2.8. Statistical Analysis

Experimental data are expressed as mean ± standard deviation (SD) and were analyzed using GraphPad Prism 9.0. Statistical analysis was performed using the two-tailed Mann–Whitney U test for non-parametric data and the unpaired *t*-test for parametric data (*p* < 0.05).

## 3. Results

### 3.1. Vaginally Exposed gEV Caused Vaginitis, Osteitis, and Neuroinflammation in Mice

Gut bacteria-producing EVs cause inflammation in vitro and in vivo [20,26]. Therefore, to understand the systemic inflammation-triggering factors of GV, we purified gEVs from the supernatant of GV culture. The gEVs were spherical vesicular structures (Figure 1, Appendix A) and consisted of proteins, LPS, and nucleic acids. Proteomic analysis using LC-MS/MS revealed that the gEVs contained type 1 polyketide synthase and a hypothetical protein (partial of *Klebsiella oxytoca*). When the gEVs were exposed to macrophages, they induced RANK, RANKL, TNF-α, and IL-6 expression dose-dependently. GV (at a dose of 10^6^ CFU/mL) induced them more potently than the gEVs (400 ng/mL).

We examined whether vaginal exposure to GV or gEVs could induce vaginitis in mice (Figure 2). GV infection caused vaginitis, increasing uterine weight and TNF-α, IL-6, and RANKL levels, along with increasing the number of NF-κB^+^ and TNF-α^+^ cells in the vagina. gEVs also led to increased uterine weight, TNF-α, IL-6, and RANKL expression, and increased numbers of NF-κB^+^ and TNF-α^+^ cells in the vagina. Moreover, gEV exposure elevated TNF-α and RANKL expression in the bloodstream, as observed in GV infection. The potency of gEVs (10 µg/mouse) in the occurrence of vaginitis was comparable to that of GV (1 × 10^6^ CFU/mouse). Although the number of TNF-α^+^ and NF-κB^+^ cells was lower in mice exposed to gEVs than in those infected with GV, the differences in their RANK, RANKL, TNF-α, and IL-6 levels were not significant.

Next, we examined whether vaginally exposed GV or gEVs could cause osteitis in mice (Figure 3). GV infection caused osteitis, leading to a notable increase in the expression of RANK, RANKL, TNF-α, IL-6, TRAP, and MMP-2 in the femur, accompanied by an increase in the number of TRAP^+^, TNF-α^+^, and RANK^+^ cells, while osteoprotegerin expression decreased. gEVs also resulted in a significant increase in TNF-α, RANK, RANKL, and IL-6 expression and the number of TRAP^+^, TNF-α^+^, and RANK^+^ cells in the femur and a significant decrease in osteoprotegerin expression. The potency of gEVs (10 µg/mouse) in the occurrence of osteitis was comparable to that of GV (1 × 10^6^ CFU/mouse). In particular, although the number of TNF-α^+^ and RANK^+^ cells was lower in mice exposed to gEVs than in mice infected with GV, the difference in RANK, RANKL, TNF-α, IL-6, TRAP, and MMP-2 expression was not significant.

The effects of GV and gEVs on the occurrence of DCi in mice were assessed (Figure 4). Vaginal exposure to GV or gEVs induced depressive-like behaviors. In the EMPT, GV and gEVs decreased the time spent in the open field (OT) by 59.8% (F_[2,15]_ = 4.070, *p* < 0.003) and 47.3% (F_[2,15]_ = 4.070, *p* = 0.023), respectively. In the OFT, they caused a reduction in the distance traveled in the central area (DC) by 41.9% (F_[2,15]_ = 12.790, *p* < 0.001) and 28.6% (F_[2,15]_ = 12.790, *p* = 0.011), respectively, and time spent in the central area (TC) by 61.2% (F_[2,15]_ = 6.392, *p* < 0.001) and 39.3% (F_[2,15]_ = 6.392, *p* < 0.001), respectively. They caused an increase in the immobility time (IT) in the TST by 39.4% (F_[2,15]_ = 6.406, *p* < 0.001) and 36.7% (F_[2,15]_ = 6.406, *p* < 0.001), respectively. They also decreased spontaneous alteration (SA) in the YMT by 41.6% (F_[2,15]_ = 15.596, *p* < 0.001) and 33.8% (F_[2,15]_ = 15.546, *p* < 0.001), respectively, and the recognition index (RI) in the NORT by 32.3% (F_[2,15]_ = 1.058, *p* < 0.001) and 26.5% (F_[2,15]_ = 1.058, *p* = 0.002), respectively. Both GV and gEVs also elevated TNF-α and IL-6 levels and NF-κB^+^Iba1^+^ cell number in the hippocampus and hypothalamus, while reducing serotonin levels. The potency of gEVs (10 µg/mouse) in the occurrence of DCi was comparable to that of GV (1 × 10^6^ CFU/mouse).

### 3.2. Vaginally Exposed gEV Deteriorated Vaginitis, Osteitis, and Neuroinflammation in Ovariectomized Mice

To understand the effects of GV and gEVs on the occurrence of systemic inflammation in menopausal women, we prepared ovariectomized mice and examined the effects of vaginally administered GV and gEVs on the expression of systemic inflammation-related biomarkers in the vagina, bloodstream, and bone (Figure 5). OV led to an increase in TNF-α, IL-6, and RANKL levels and NF-κB^+^ and TNF-α^+^ cell number in the vagina, while reducing uterine weight. When GV or gEVs were vaginally applied to mice with OV, they significantly enhanced uterine weight and TNF-α, IL-6, and RANKL levels, along with increasing the TNF-α^+^ and NF-κB^+^ cell number. In the blood, OV increased RANKL levels and reduced estradiol levels. Exposure to GV or gEVs further amplified OV-induced TNF-α and RANKL levels, although they decreased the OV-reduced estradiol level. The potency of gEVs (10 µg/mouse) in the occurrence of vaginitis was comparable to that of GV (1 × 10^6^ CFU/mouse) in mice with OV. In particular, the number of TNF-α^+^ and RANK^+^ cells was lower in mice exposed to gEVs than in mice infected with GV. Estradiol levels significantly decreased in mice exposed to gEVs alone. However, the differences in their uterine weight, TNF-α, IL-6, and RANKL levels were not significant.

OV also decreased femur weight and osteoprotegerin expression in the femur while increasing TNF-α, RANK, RANKL, and IL-6 levels and TRAP^+^, RANK^+^, and TNF-α^+^ cell number (Figure 6). Vaginally exposed GV or gEVs exacerbated the OV-induced elevation in RANK, RANKL, TNF-α, and IL-6 levels and in TRAP^+^, RANK^+^, and TNF-α^+^ cell populations, while further reducing femur weight and osteoprotegerin level. The potency of gEVs (10 µg/mouse) in the occurrence of osteitis was comparable to that of GV (1 × 10^6^ CFU/mouse) in mice with OV. In particular, although RANK and RANKL expression were higher in mice exposed to gEVs than in mice infected with GV, and the osteoprotegerin level was lower in mice exposed to gEVs than in mice infected with GV, the differences in their uterine weight, TNF-α, and IL-6 levels were not significant.

We also investigated whether vaginally exposed GV or gEVs could cause DCi in ovariectomized mice (Figure 7). OV significantly induced severe DCi-like behaviors compared to sham mice (Sh). Vaginal exposed GV or gEVs showed more severe DCi-like behaviors in addition to DCi-like behavior caused by OV. In particular, compared to Sh mice treated with vehicle (saline) alone, the exposure of mice with OV to gEVs reduced DC to 15.3% (F_[3,20]_ = 8.055, *p* < 0.001), TC to 13.4% (F_[3,20]_ = 7.914, *p* < 0.001), and OT to 9.2% (F_[3,20]_ = 13.29, *p* < 0.001), and increased IT to 187.1% (F_[3,20]_ = 2.116, *p* < 0.001). gEVs also decreased SA to 66.3% (F_[3,20]_ = 1.433, *p* < 0.001) and RI to 56.5% (F_[3,20]_ = 3.859, *p* < 0.001). Compared to Ov group mice treated with vehicle alone, vaginal exposure of Ov group mice to gEVs reduced DC to 62.1% (F_[3,20]_ = 26.431, *p* < 0.001), TC to 62.7% (F_[3,20]_ = 23.182, *p* < 0.001), and OT to 28.2% (F_[3,20]_ = 57.403, *p* < 0.001), and increased IT to 122.9% (F_[3,20]_ = 221.021, *p* < 0.001). In the Ov group, gEVs also decreased SA to 90.7% (F_[3,20]_ = 19.110, *p* < 0.001) and RI to 85.4% (F_[3,20]_ = 73.04, *p* < 0.001). Exposure to gEVs or GV also increased TNF-α and IL-6 levels in the hippocampus and hypothalamus, while serotonin decreased. In particular, exposure to GV or gEVs significantly caused DCi-like behaviors in the TST and RI in the NORT compared to those in Ov. And they significantly increased TNF-α and IL-6 levels in the hippocampus and TNF-α levels in the hypothalamus, while serotonin levels decreased in the hippocampus and hypothalamus. However, the differences in their depression-related parameters—except IT in the TST and serotonin levels in the hypothalamus—were not significant.

### 3.3. FITC-Labeled gEV Administered Vaginally Is Transported into the Bloodstream, Hippocampus, and Femur

To investigate whether vaginally exposed gEVs could be delivered to systemic organs including the brain and femur, we prepared FITC-tagged gEVs and exposed the vagina to them (Figure 8). The FITC-labeled gEvs were observed in the blood, hippocampus, and femur.

## 4. Discussion

Vaginally infected GV leads to vaginitis by activating the NF-κB signaling pathway [3,27]. We also observed that vaginal infection with GV results in elevated levels of TNF-α, IL-6, and RANKL, as well as an elevated number of NF-κB-positive cells in the vagina, femur, hippocampus, and hypothalamus, leading to vaginitis, osteitis, and neuroinflammation, as previously reported [22]. Furthermore, GV infection reduced hippocampal serotonin levels and increased DCi-like behaviors in mice with or without OV. Elevated TNF-α levels in postmenopausal women are known to accelerate RANKL-induced osteoclast formation, contributing to osteoporosis [15,16]. TNF-α and RANKL promote osteoclast differentiation by activating RANK-involved NF-κB signaling, ultimately leading to osteoporosis [16,28,29]. Bacterial vaginosis may be linked to diminished emotional and social health [30]. Patients with osteoporosis have a higher risk of depression than those without osteoporosis [31]. Anti-RANKL antibody (Denosumab) can alleviate both osteoporosis and depression [32]. Elevated RANK levels in the bloodstream are associated with depression symptoms [33]. Collectively, these findings suggest that GV infection may contribute to comorbid conditions, such as DCi and osteoporosis, through the activation of RANK/RANKL-mediated NF-κB signaling.

Recent emerging research highlights that EVs produced by gut pathogens, such as *Paenalcaligenes hominis* and *Escherichia fergusonii*, play a role in systemic diseases, including cognitive dysfunction [20,21]. Similarly, EVs from *Porphyromonas gingivalis* can travel from the oral cavity to the brain through the trigeminal nerve and blood circulation, leading to neuroinflammation [19]. In the present study, vaginal exposure to gEVs significantly caused DCi-like behaviors in mice with or without OV, similarly to GV-infected mice. gEVs also increased TNF-α, IL-6, and RANKL levels in the bloodstream of mice with or without OV, and increased RANK, RANKL, and TNF-α expression and NF-κB-positive cell population in the vagina, femur, hippocampus, and hypothalamus, while serotonin levels decreased in the hippocampus and hypothalamus. These results suggest that vaginally exposed gEVs may cause vaginitis, osteitis, and neuroinflammation through the activation of RANK/RANKL-mediated NF-κB signaling, resulting in DCi with systemic inflammation.

gEVs consisted of LPS, proteins, and nucleic acids. Of proteins, type 1 polyketide synthase, which may contribute to polyketide toxin production [34], and a hypothetical protein from *Klebsiella oxytoca*, a cytolytic bacterium [35], were contained in gEVs, as assessed by proteomic analysis. LPS is a well-known inflammation-inducer in vitro and in vivo [24,36]. For the first time, we found that vaginally exposed gEVs caused vaginitis, osteitis, and neuroinflammation through the activation of NF-κB. Vaginally infected GV causes vaginitis, osteitis, colitis, and neuroinflammation [22]. Gut dysbiosis-induced gut inflammation causes systemic inflammation, including neuroinflammation, leading to psychiatric disorders through the gut–brain axis [37,38]. These results suggest that gEVs may be a triggering ingredient for systemic inflammation in GV through the vagina–brain axis and/or vagina–gut–brain axis.

When FITC-labeled gEVs were introduced into the vagina, they were detected in the blood, femur, and hippocampus. These observations suggest that vaginally exposed gEVs can be highly pathogenic, potentially spreading to systemic organs including the brain and bones, inducing inflammation via NF-κB signaling, leading to vaginitis, osteoporosis, and DC, as systemic inflammation caused by GV. To fully elucidate the pathogenesis of GV, further research may be essential to identify the route by which gEVs are delivered to the blood, bone, and brain.

## 5. Conclusions

Vaginal exposure to gEVs increased the expression of proinflammatory cytokines including TNF-α and RANKL in mice with or without OV by activating NF-κB signaling, similar to the effect of GV infection, leading to vaginitis (Figure 1). gEV-induced vaginal inflammation may trigger osteitis and neuroinflammation through RANK/RANKL-mediated NF-κB activation, resulting in osteoporosis and DCi. Therefore, gEVs may be an important risk factor for vaginitis, as well as osteoporosis and neuropsychiatric disorders in women.

## Data Availability

The original contributions presented in this study are included in the article/Appendix A. Further inquiries can be directed to the corresponding author.

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
