# Peer review of "The Vaginally Exposed Extracellular Vesicle of Gardnerella vaginalis Induces RANK/RANKL-Involved Systemic Inflammation in Mice"

_microorganisms, 2025, doi:10.3390/microorganisms13040955_

Round 1
Reviewer 1 Report
Comments and Suggestions for Authors
This study requires extensive revision before publication in the Microorganisms Journal. The comments are provided below as follows:
- Abstract: The aim of this study should be clearly described. Additionally, key findings should be highlighted, including quantitative data to strengthen the abstract.
- Intro: It is too short and not suitable for the comprehensive results presented in this study. The authors should also justify the novelty of this study. The objective of this study should be clearly discussed. Generally, this section should be rewritten.
- Methods: the authors are encouraged to add a scheme, showing the flow of this study and animal groups to avoid confusion for readers. Line 56: as previously described. Line 74: The kits used in this study should be mentioned (name of the kit, Cat. No., city, and country). The authors should elaborate on the methods applied in this study for reproducibility. Line 111: Please add details of approaches and kits as previously mentioned. Statistical analysis: Did you normalize your data?
- Results: All figures are not clear. Please enhance the quality of the figures. Fig. 2e: scale bars are invisible. Please consider a similar comment for Fig. 3H, 5e, and 8.
- Discussion: the novelty and aim of this study should firstly be discussed to present a cohesive story of this study. It seems that the authors did not compare their findings with previous studies and they stated some of the previous findings. In light of the findings shown in this study, the authors should propose the potential mechanism and add a figure for this mechanism. The limitations of this study should be discussed.
- Conclusion: The authors should highlight the key findings and recommendations.
The authors should scrutinize the manuscript.
Author Response
This study requires extensive revision before publication in the Microorganisms Journal. The comments are provided below as follows:
- Abstract: The aim of this study should be clearly described. Additionally, key findings should be highlighted, including quantitative data to strengthen the abstract.
-->Thank you. We wrote the abstract again according to your comment. Line 10-33.
- Intro: It is too short and not suitable for the comprehensive results presented in this study. The authors should also justify the novelty of this study. The objective of this study should be clearly discussed. Generally, this section should be rewritten.
-->Thank you. We significantly revised the introduction section according to your comment. Line 38-39, Line 42, Line 46-53, Line 57-58.
- Methods: the authors are encouraged to add a scheme, showing the flow of this study and animal groups to avoid confusion for readers. Line 56: as previously described. Line 74: The kits used in this study should be mentioned (name of the kit, Cat. No., city, and country). The authors should elaborate on the methods applied in this study for reproducibility. Line 111: Please add details of approaches and kits as previously mentioned. Statistical analysis: Did you normalize your data?
-->Thank you. We partially described experimental protocols. And we added reagents-related company, city, and country and assay kits-related company. qPCR’s data were calculated, compared to those of normal mice (NC).
- Results: All figures are not clear. Please enhance the quality of the figures. Fig. 2e: scale bars are invisible. Please consider a similar comment for Fig. 3H, 5e, and 8.
-->Thank you. We significantly revised them. The size scale bars were indicated in Figure 2e. 3h, 5e.
- Discussion: the novelty and aim of this study should firstly be discussed to present a cohesive story of this study. It seems that the authors did not compare their findings with previous studies and they stated some of the previous findings. In light of the findings shown in this study, the authors should propose the potential mechanism and add a figure for this mechanism. The limitations of this study should be discussed.
-->Thank you. We rewrote discussion section. Line 337-340, Line 357-376, Line 382-384.
- Conclusion: The authors should highlight the key findings and recommendations.
-->Thank you. we rewrote conclusions section and added scheme 1.
Reviewer 2 Report
Comments and Suggestions for Authors
In this manuscript“The vaginally exposed extracellular vesicle of Gardnerella vaginalis induces RANK/RANKL-involved systemic inflammation in mice”, The author outlines that peripheral inflammation is closely related to the occurrence of systemic lupus erythematosus. Understand the pathogenic factors of GV in systemic inflammation and the pathogenic role of GV (GeV) extracellular vesicles in mice. GEV exposure can also lead to depression/cognitive impairment (DC)-like behaviors. In addition, the author's research has found that GV-derived GeVs may induce systemic inflammation by activating GV-GV, and RANK/RANKL is involved in NF -kB signal transduction, resulting in systemic diseases including the vagus nerve, otitis, osteoporosis and DC, etc. However, there had some problems and questions, the comments were as followed:
- It is recommended that the authors add a quantitative analysis to the abstract section.
- The authors' introductory section is not descriptive and logical and would like to re-describe it. For example, introduce systemic inflammation. Then describe it in a recursive relationship.
- There should be a space between the number and the unit.
- “Fluorescein isothiocyanate (FITC)-labeled gEV (F-gEV) was prepared, as previously reported.” suggest to add specific method to describe it.
- “Exposure of mice to GV or gEV” to check if there are any problems.
- Why was systemic inflammation not considered to link the colon to the brain, but instead chose to do only the vagina, uterus, and brain?
- The discussion section of the article does not explain how this article differs from existing articles, and it is recommended that the authors add a description of this section.
- What is the problem orientation of this article? What is the point of innovation? How does it differ from existing literature.
- What are the limitations of this article? It is recommended that the authors add this part of the description.
- The quality of the images could also be improved.
Author Response
In this manuscript“The vaginally exposed extracellular vesicle of Gardnerella vaginalis induces RANK/RANKL-involved systemic inflammation in mice”, The author outlines that peripheral inflammation is closely related to the occurrence of systemic lupus erythematosus. Understand the pathogenic factors of GV in systemic inflammation and the pathogenic role of GV (GeV) extracellular vesicles in mice. GEV exposure can also lead to depression/cognitive impairment (DC)-like behaviors. In addition, the author's research has found that GV-derived GeVs may induce systemic inflammation by activating GV-GV, and RANK/RANKL is involved in NF -kB signal transduction, resulting in systemic diseases including the vagus nerve, otitis, osteoporosis and DC, etc. However, there had some problems and questions, the comments were as followed:
- It is recommended that the authors add a quantitative analysis to the abstract section.
-->Thank you. We wrote the abstract again according to your comment. Line 10-33.
- The authors' introductory section is not descriptive and logical and would like to re-describe it. For example, introduce systemic inflammation. Then describe it in a recursive relationship.
-->Thank you. We significantly revised the introduction section according to your comment. Line 38-39, Line 46-53, Line 57-58.
- There should be a space between the number and the unit.
-->Thank you. We revised it according to your comment.
- “Fluorescein isothiocyanate (FITC)-labeled gEV (F-gEV) was prepared, as previously reported.” suggest to add specific method to describe it.
-->Thank you. We revised it according to your comment. Line 93-94.
- “Exposure of mice to GV or gEV” to check if there are any problems.
-->Thank you. We revised it according to your comment. Line 113-114, Line 120, Line 123-124.
- Why was systemic inflammation not considered to link the colon to the brain, but instead chose to do only the vagina, uterus, and brain?
-->Thank you. We added its related discussion in Discussion section. We reported that the vaginal infection of GV caused vaginitis as well as colitis in the previous study (ref 22). Therefore, we suggest that GV, particularly, cause systemic inflammation, leading to depression through (1) the axis of vagina to brain and (2) the axis of the vagina to brain through intestine. Line 370-376.
- The discussion section of the article does not explain how this article differs from existing articles, and it is recommended that the authors add a description of this section.
-->Thank you. We added its related discussion in Discussion section. Line 357-382.
- What is the problem orientation of this article? What is the point of innovation? How does it differ from existing literature.
--> Thank you. We added its related discussion in Discussion section Line 370-376.
- What are the limitations of this article? It is recommended that the authors add this part of the description.
-->Thank you. We added it. Line 382-384.
- The quality of the images could also be improved.
-->Thank you. we changed all figures to Figures with high resolution.
Reviewer 3 Report
Comments and Suggestions for Authors
The manuscript entitled “The vaginally exposed extracellular vesicle of Gardnerella vaginalis induces RANK/RANKL-involved systemic inflammation in mice” investigates the pathogenic role of extracellular vesicles (gEV) isolated from Gardnerella vaginalis (GV) in ovariectomized and non-ovariectomized mice. The study evaluates the expression of RANK, RANKL, TNF-α, and the number of NF-κB+ cells in the vagina, femur, hypothalamus, and hippocampus. Overall, a major revision is suggested. Please refer to the specific comments below:
- The introduction section is brief and imprecise, lacking a detailed description of the characteristics of Gardnerella vaginalis and bacterial extracellular vesicles (EVs). Furthermore, I strongly recommend including a detailed explanation of the RANK/RANKL signaling pathway and a schematic illustration to better clarify the proposed mechanism of action and the related intracellular signaling cascade.
- Lines 28-30: The sentence “Pathogen-induced inflammation can contribute to the higher prevalence of depression in women, who are diagnosed at twice the frequency of men be a cause of depression strongly linked to depression” is unclear and grammatically incorrect. I recommend rephrasing it for clarity and to avoid redundancy.
- The Materials and Methods section is insufficient and superficial. Each subparagraph is overly short, lacking essential details necessary to understand the experiments fully.
- In subparagraph 2.1, the methods for the isolation and the characterization of extracellular vesicles are not sufficiently clear. Expand this section to provide a detailed description of how each experiment was conducted. Furthermore, performing Nanoparticle Tracking Analysis (NTA) would be valuable for determining the concentration and size distribution of the particles isolated from GV.
- In subparagraph 2.3 (lines 75-76), it is necessary to explain how RNA was extracted from the macrophage cell line.
- In subparagraph 2.4 (line 81), it is required to explain the methodology used to treat the mice's vagina with bacteria or extracellular vesicles.
- In subparagraph 2.4 (Experiment 1 - line 85) and (Experiment 2 – line 89) the differences between the various experimental groups are not specified. I recommend that the authors provide more details to clarify the experimental design.
- In subparagraph 2.5, Although references are provided, I recommend providing a brief description of each behavioral test (OFT, EPMT, TST, YMT, NORT) to ensure the manuscript is accessible even to readers who may not be fully familiar with these assays.
- In paragraph 2.7, it is necessary to explain in detail the steps of immunofluorescence, starting from the sample preparation.
- Figure 1 (subparagraph 3.1) shows an image of the EVs obtained by Transmission Electron Microscopy (TEM). The Materials and Methods section should include a description of TEM procedure, including details on grid preparation.
- The results are inaccurate and difficult to interpret. I strongly recommend revising them to improve the clarity, including a reorganization into multiple subparagraphs to facilitate the interpretation of each experiment.
- Line 132: The relevance of the “EVs purified from gut bacteria P. hominis and E. fergusonii” is unclear, as this paragraph introduces them. I recommend providing further clarification. Additionally, the full names of the bacteria species should be written in full the first time they are mentioned in the text.
- Line 135: I suggest using the term “spherical vesicular structures “instead of “round, vesicular structure” for improved scientific accuracy.
Comments on the Quality of English Language
English language should be revised throughout the manuscript.
Author Response
The manuscript entitled “The vaginally exposed extracellular vesicle of Gardnerella vaginalis induces RANK/RANKL-involved systemic inflammation in mice” investigates the pathogenic role of extracellular vesicles (gEV) isolated from Gardnerella vaginalis (GV) in ovariectomized and non-ovariectomized mice. The study evaluates the expression of RANK, RANKL, TNF-α, and the number of NF-κB+ cells in the vagina, femur, hypothalamus, and hippocampus. Overall, a major revision is suggested. Please refer to the specific comments below:
The introduction section is brief and imprecise, lacking a detailed description of the characteristics of Gardnerella vaginalis and bacterial extracellular vesicles (EVs). Furthermore, I strongly recommend including a detailed explanation of the RANK/RANKL signaling pathway and a schematic illustration to better clarify the proposed mechanism of action and the related intracellular signaling cascade.
-->Thank you. We revised introduction section according to your comment. Line 38-39, Line 42, Line 46-58.
Lines 28-30: The sentence “Pathogen-induced inflammation can contribute to the higher prevalence of depression in women, who are diagnosed at twice the frequency of men be a cause of depression strongly linked to depression” is unclear and grammatically incorrect. I recommend rephrasing it for clarity and to avoid redundancy.
-->Thank you. We revised it. Line 48-49.
The Materials and Methods section is insufficient and superficial. Each subparagraph is overly short, lacking essential details necessary to understand the experiments fully.
-->Thank you. We significantly revised Materials and Methods section. Line 77-94,
Line 156-163.
In subparagraph 2.1, the methods for the isolation and the characterization of extracellular vesicles are not sufficiently clear. Expand this section to provide a detailed description of how each experiment was conducted. Furthermore, performing Nanoparticle Tracking Analysis (NTA) would be valuable for determining the concentration and size distribution of the particles isolated from GV.
--> Thank you. We significantly revised Materials and Methods section. Line 77-93,
Line 86-93.
In subparagraph 2.3 (lines 75-76), it is necessary to explain how RNA was extracted from the macrophage cell line.
--> Thank you. We significantly revised Materials and Methods section. Line 87-89.
In subparagraph 2.4 (line 81), it is required to explain the methodology used to treat the mice's vagina with bacteria or extracellular vesicles.
--> Thank you. We significantly revised Materials and Methods section. Line 78-86, Line 87-94.
In subparagraph 2.4 (Experiment 1 - line 85) and (Experiment 2 – line 89) the differences between the various experimental groups are not specified. I recommend that the authors provide more details to clarify the experimental design.
-->Thank you. We revised the section. Line 113-132.
In subparagraph 2.5, Although references are provided, I recommend providing a brief description of each behavioral test (OFT, EPMT, TST, YMT, NORT) to ensure the manuscript is accessible even to readers who may not be fully familiar with these assays.
-->Thank you. We fully described it in Supplement Section.
In paragraph 2.7, it is necessary to explain in detail the steps of immunofluorescence, starting from the sample preparation.
-->Thank you. We added it. Line 157-164.
Figure 1 (subparagraph 3.1) shows an image of the EVs obtained by Transmission Electron Microscopy (TEM). The Materials and Methods section should include a description of TEM procedure, including details on grid preparation.
-->Thank you. We added it. Line 83-87.
The results are inaccurate and difficult to interpret. I strongly recommend revising them to improve the clarity, including a reorganization into multiple subparagraphs to facilitate the interpretation of each experiment.
-->Thank you. We significantly revised result section according to your comment. Line 179-180, Line 185-186, Line 195, Lime 200-204, Line 219-223. Line 244-245, Line 266-270, Line 283-288. Line 304-315,
Line 132: The relevance of the “EVs purified from gut bacteria P. hominis and E. fergusonii” is unclear, as this paragraph introduces them. I recommend providing further clarification. Additionally, the full names of the bacteria species should be written in full the first time they are mentioned in the text.
-->Thank you. We revised it. Line 59-62.
Line 135: I suggest using the term “spherical vesicular structures “instead of “round, vesicular structure” for improved scientific accuracy.
--> Thank you. We revised it. Line 180.
Round 2
Reviewer 1 Report
Comments and Suggestions for Authors
The authors did a great job addressing the previous claims. However, one comment should still be addressed. The author did not refer to scheme 1 in the context of the manuscript.
Author Response
The authors did a great job addressing the previous claims. However, one comment should still be addressed. The author did not refer to scheme 1 in the context of the manuscript.
-->Thank you. We referred it in Line 389. And we revised some errors.
Reviewer 3 Report
Comments and Suggestions for Authors
The authors have addressed all queries satisfactorily. The article is suitable for publication in its current form.
Author Response
The authors have addressed all queries satisfactorily. The article is suitable for publication in its current form.
-->Thank you.